# Atomic structure and domain wall pinning in samarium-cobalt-based permanent magnets

M. Duerrschnabel[1], M. Yi[2], K. Uestuener[3], M. Liesegang[3,4], M. Katter[3], H.-J. Kleebe[1], B. Xu[2], O. Gutfleisch[4] & L. Molina-Luna[1]

A higher saturation magnetization obtained by an increased iron content is essential for yielding larger energy products in rare-earth $Sm_2Co_{17}$-type pinning-controlled permanent magnets. These are of importance for high-temperature industrial applications due to their intrinsic corrosion resistance and temperature stability. Here we present model magnets with an increased iron content based on a unique nanostructure and -chemical modification route using Fe, Cu, and Zr as dopants. The iron content controls the formation of a diamond-shaped cellular structure that dominates the density and strength of the domain wall pinning sites and thus the coercivity. Using ultra-high-resolution experimental and theoretical methods, we revealed the atomic structure of the single phases present and established a direct correlation to the macroscopic magnetic properties. With further development, this knowledge can be applied to produce samarium cobalt permanent magnets with improved magnetic performance.

[1] Department of Material- and Geosciences, Technische Universität Darmstadt, Alarich-Weiß-Strasse 2, Darmstadt D-64287, Germany. [2] Department of Material- and Geosciences, Mechanics of Functional Materials Division, Technische Universität Darmstadt, Jovanka-Bontschits-Strasse 2, Darmstadt D-64287, Germany. [3] Vacuumschmelze GmbH & Co. KG, Grüner Weg 37, Hanau D-63450, Germany. [4] Department of Material- and Geosciences, Functional Materials, Technische Universität Darmstadt, Alarich-Weiß-Str. 16, Darmstadt D-64287, Germany. Correspondence and requests for materials should be addressed to L.M-L. (email: molina@geo.tu-darmstadt.de)

Pinning-controlled permanent magnets operating at elevated temperatures boost device performances of magnet-based industrial applications[1–9]. These include microwave tubes, gyroscopes and accelerometers, reaction and momentum wheels to control and stabilize satellites, magnetic bearings, sensors and actuators. $Sm_2(Co,Fe,Cu,Zr)_{17}$ is an important industrially used material system, since it has both a high Curie temperature and a high magnetocrystalline anisotropy[10–14]. Unlike nucleation-controlled Nd-Fe-B-based permanent magnets, the $Sm_2Co_{17}$-type maintains its excellent magnetic properties at elevated temperatures[15]. In order to precisely control the synthesis parameters to obtain such high magnetic performances, it is necessary to thoroughly understand the atomic-scale structure and behavior of the involved phases. This is not a straightforward task and although the relationship of micro-structure and chemistry with the magnetic properties has been widely studied by local techniques such as electron microscopy, the number of atomic-scale investigations is still limited[1–3, 16–20].

The iron content has a significant effect on the magnetic properties of $Sm_2(Co,Fe,Cu,Zr)_{17}$ permanent magnets[21–26]. It was shown by Hadjipanayis et al.[4] that an optimum coercivity is reached for an iron content between 15 and 20 wt%. With increasing Fe content, the cellular structure changes from an inhomogeneous to a larger, but uniform cell size (~120 nm), and finally to a coarse and inhomogeneous microstructure[27]. Iron preferentially replaces cobalt in the 2:17 phase and is responsible for the saturation magnetization. Since the domain wall energy is largest in the cell boundary phase (later referred as $SmCo_5$ or 1:5 phase), this phase acts as a main pinning center for magnetic domain walls[21, 28]. According to Skomski et al.[21] Zr-rich (Z-phase) platelets contribute to the formation of the cell boundaries and do not yield any dominating contribution to the coercivity, but might still act as pinning centers. Skomski et al.[21] as well as Katter et al.[29, 30] predicted that the domain walls are heavily bowed until they reach an interface between the 2:17 and the 1:5 phase. However, the pinning forces at such straight interfaces are much higher than the observed coercivities. Therefore, the coercivity is determined by the depinning of the domain walls at certain weak points[31]. These weak points are the edges of the 2:17 cells and the intersection lines of the 1:5 phase with the Z-phase. The domain walls are strongly pinned at the plane interfaces between the 2:17 cell and the 1:5 boundary phase.

In the following, we present a detailed investigation on the atomic scale of the Z-phase and its contribution to the domain wall-pinning behavior. We demonstrate that it is much favorable from an energetic point of view to move a short section of the domain wall at these weak points from the 2:17 or Z-phase into the 1:5 phase than to press a long section of it into the plane interface. This implies that the domain walls are not only pinned at the plane 1:5 to 2:17 interface, but are also firstly depinned at the edges of the cells and later at the intersection lines of the 1:5 and the Z-phase. In order to clarify the atomic structure of the Z-phase and its contribution to the magnetization process, we investigated in detail the microstructure by combined atomic-structure investigations, microstructure-based micromagnetic simulations and density functional theory calculations.

## Results

**Transmission electron microscopy.** Figure 1 shows bright-field transmission electron microscopy (TEM) images and selected area electron diffraction (SAED) patterns of two different samples. Figure 1a is a bright-field TEM image of sample 1 (lower iron content, see Table 1) oriented close to the [110] pole in two-beam condition. The diamond-shaped cellular structure of the 1:5 boundary phase and the Z-phase, therefore show strong

diffraction contrast. A detailed energy-dispersive X-ray micro-analysis (EDX) of the single phases is presented in Supplementary Fig. 1 and Supplementary Table 1. The diamond-shaped cellular structure has a uniform size ~200 nm and is very well aligned. The Z-phase platelets are ~4 nm in height and are densely distributed. They intersect the 10-nm-thick diamond-shaped 1:5 boundary phase. Figure 1b shows a SAED pattern along the [110] zone-axis (red part) with a [100] oriented twin (blue part). The inset shows a line profile along the [00l] direction. The additional reflections marked by the triangles as well as the slight streaking originate from the Z-phase platelets forming an ordered super-structure along the c-axis direction (see the {0,0,3/2} type of reflections revealed by the line profile). Figure 1c is a bright-field TEM image of sample 2 (higher iron content, see Table 1) oriented close to the [210] pole in two-beam condition. The Z-phase shows here diffraction contrast, however, the striking difference compared to sample 1 is that there is no diamond-shaped cellular structure in the 1:5 boundary phase present. Only single, isolated facets of 1:5 cells are found. Figure 1d shows a SAED along the [210] zone-axis. The different zone-axis orientation makes no difference to the visibility of the 1:5 cellular structure in the two-beam condition obtained bright-field TEM images. The strong streaking being present in the SAED along the [00l] direction originates from the presence of the Z-phase platelets. The inset shows a line profile along the [00l] direction with additional reflections marked by triangles. These additional reflections do not lie on the center between two regular spots of the $Sm_2Co_{17}$ (2:17) phase as it is the case for sample 1 indicating together with the stronger streaking a larger Z-phase disorder for sample 2 as compared to sample 1. Therefore, it is obvious that the structure of sample 2 strongly deviates from that of sample 1: There is no diamond-shaped cellular structure of the 1:5 phase visible at all. However, still isolated unequally distributed 1:5-type lamellas occur. Some of them seem to act as bridges perpendicularly connecting to the Z-phase platelets. Those bridges may be boundaries between two phases, as the Z-phase platelets should only grow in one direction. A few of the Z-phase platelets also suddenly end somewhere in the 2:17 matrix, especially in sample 2.

**Scanning transmission electron microscopy.** An ultra-high resolution scanning TEM high-angle annular dark-field (STEM-HAADF) Z-contrast image is shown in Fig. 2a to reveal the atomic structure of the Z-phase. We implemented two models, $SmCo_3$ and $Zr_2SmCo_9$ for the QSTEM simulations as shown in Fig. 2b[32]. The later one is a modification of the first one, where the Sm at the Sm1 (6c) atomic position was replaced by Zr. The models were chosen since it was not clear from the beginning which atomic position the Zr would occupy and if there is only a partial or a full replacement of Sm by Zr on the Sm1 (6c) position. A simulated STEM-HAADF Z-contrast image of pure $SmCo_3$ is shown in Fig. 2c together with an atomic model inside the Z-phase. The same is shown for the Zr-modified structure in Fig. 2d. One directly recognizes that the intensity of the Sm1 (6c) position in Fig. 2c is too bright in the simulation compared to the experiment (Fig. 2a), because Sm ($Z = 62$) is heavier than Zr ($Z = 40$). Therefore, a simulation with the modified model was carried out yielding the results presented in Fig. 2d. By comparing the simulated image with the experimental one, a perfect match was found regarding the atomic intensities within both images showing that only the Sm1 (6c) position is replaced by Zr. Indications of this behavior were suggested by X-ray measure-ments before[33, 34]. The Sm2 (3a) position is still occupied by a Sm atom. By estimating the site-preference energy via first principle calculations (Supplementary Fig. 5), we found that when one or

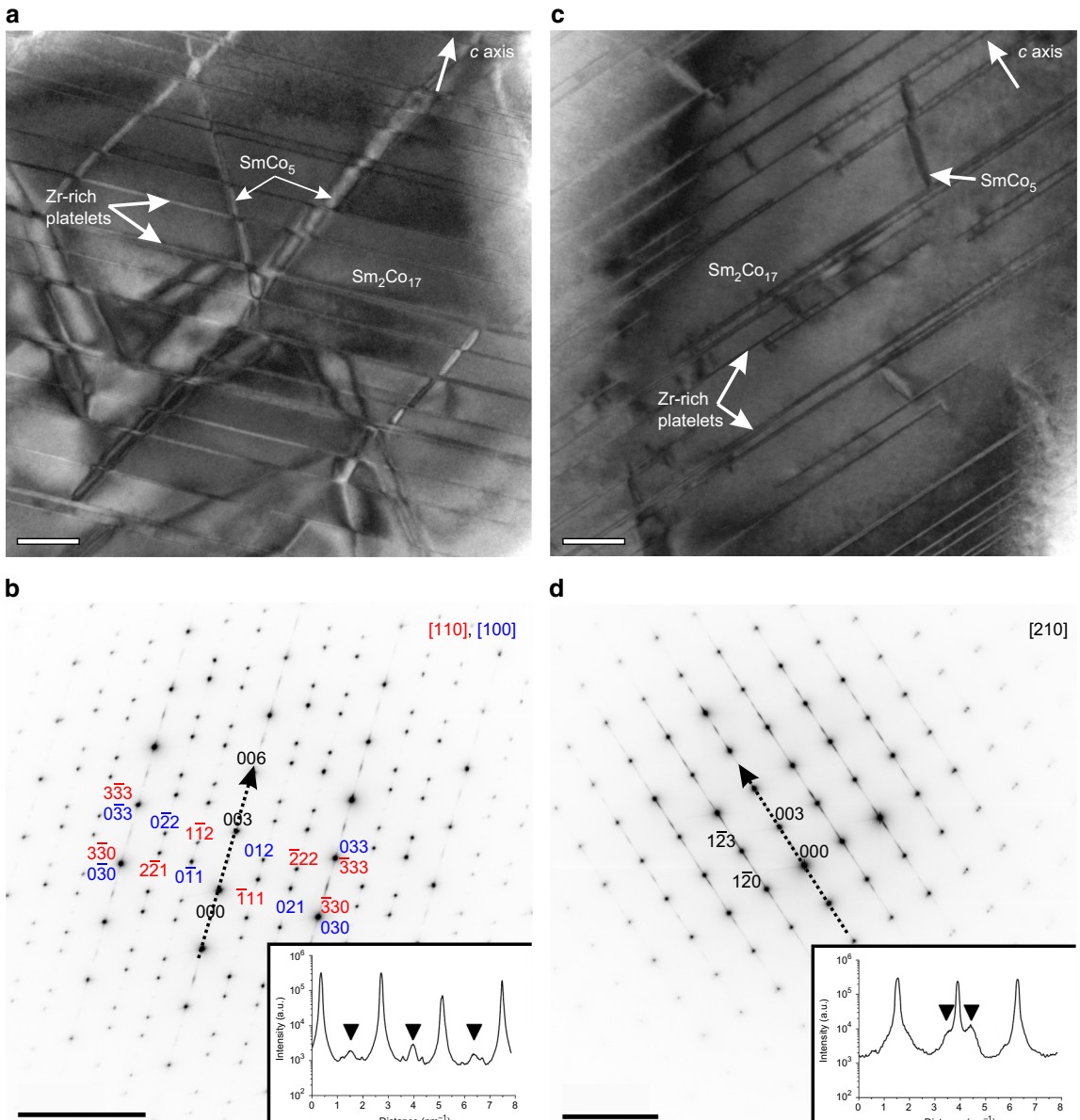

**Fig. 1** Nanoscale phase distribution. **a** Bright-field TEM image of sample 1. **b** Corresponding selected area electron diffraction pattern along the [110] zone axis (red text labels) with additional reflections from a [100] oriented twin (blue text labels). The superstructure reflections of type {0,0,3/2} along the hexagonal c-axis are denoted in the line profile shown in the inset of **b**. **c** Bright-field TEM image of sample 2. **d** Corresponding selected area diffraction pattern along the [210] zone-axis. Note the difference in ordering in the line profile inset in **d** compared to the line profile shown in the inset in **b**. The scales bars in the TEM images correspond to 50 nm. The scale bars in the electron diffraction patterns correspond to 5 nm$^{-1}$

**Table 1 Iron content and magnetic properties of the samples**

| Sample | Nominal Fe content (wt%) | $B_r$ (T) | $H_{cB}$ (kA m$^{-1}$) | $H_{cJ}$ (kA m$^{-1}$) | $(BH)_{max}$ (kJ m$^{-3}$) |
|---|---|---|---|---|---|
| 1 | 19 | 1.2 | 870 | 2,380 | 262 |
| 2 | 23 | 0.9 | 250 | 280 | 100 |

The data were extracted from demagnetization curves obtained at $T = 20\,°C$. The determined quantities are remanence ($B_r$), coercive field strength at polarization equals zero ($H_{cB}$), coercive field strength at flux density equals zero ($H_{cJ}$) and energy density (($BH$)$_{max}$). These values can be compared to the values provided by Maybury et al.[15]

all Sm (6c or 3a) sites are occupied with Zr atoms, the Sm1 (6c) site has a stronger preference to be occupied by Zr, since it is energetically favorable.

The Z-phase itself has a layered structure (Z-phase stacks) and is in some cases inhomogeneous, that is, contains for example stacking faults, as shown in Supplementary Fig. 2. Supplementary

Fig. 2 shows STEM-HAADF Z-contrast images of different Z-phase stacks oriented along the [120] zone axis. The 2:17 matrix is oriented along the [110] or [100] zone axis. Supplementary Fig. 2a and b shows defect-free Z-phase stacks as indicated by numbered yellow arrows with two and four stacks, respectively. These defect-free Z-phase stacks are more likely

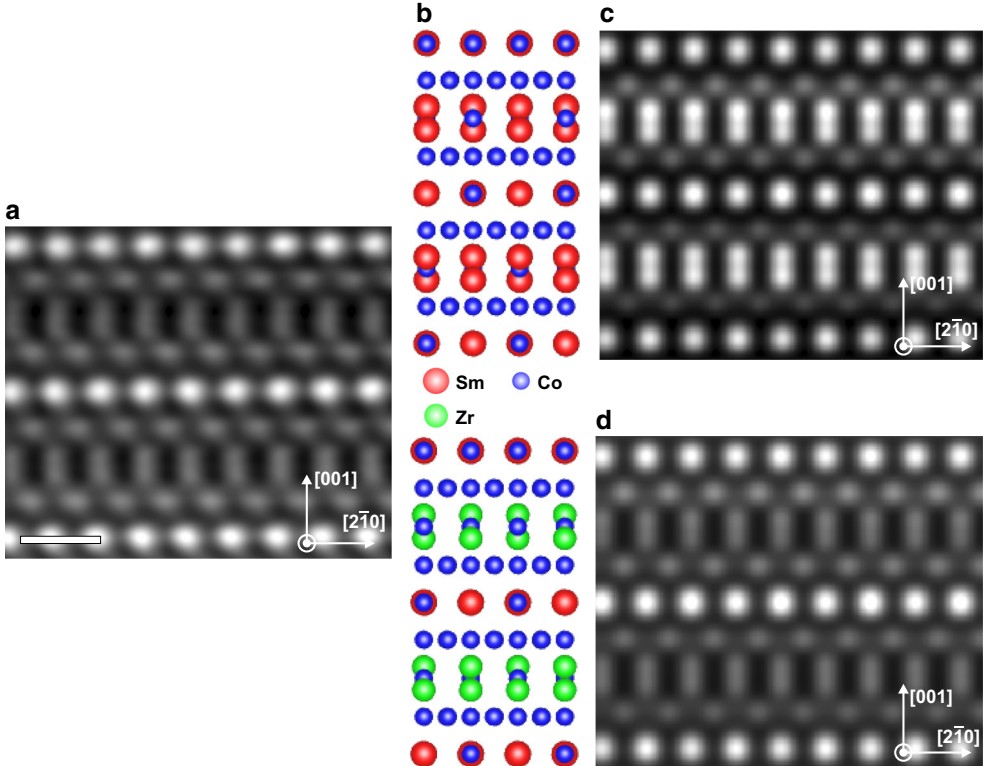

**Fig. 2** Atomic-resolution HAADF-STEM images of the Z-phase. **a** Experimental (*left*), **b** atomic models (*center*) and simulated (*right*) atomic resolution STEM-HAADF Z-contrast images for **c** SmCo₃ and **d** Zr₂SmCo₉. All images are viewed along the [120] zone axis. The experimental image was filtered by principal component analysis to reduce the effect of noise. Scale bar, 5 Å

being observed in the low iron content sample. Supplementary Fig. 2c and d is quadruple and sextuple Z-phase stacks containing stacking faults. These stacking faults consist of one or more Sm/Co-Co layers. The defective Z-phase stacks are more likely found in the high iron content sample. This structural feature has not been reported before and shows that the diffusion of elements like Cu and Fe during the annealing step is suppressed for the high iron content sample.

Analyzing Supplementary Fig. 3 in detail demonstrates that three twinning structures can occur in Sm₂Co₁₇-based permanent magnets. Feng et al.[3] proposed structural models regarding the 2:17 matrix and the 1:5 boundary phase suggesting how the twinning should look like on the atomic scale by systematically evaluating electron diffraction patterns. Their findings are confirmed also on the atomic scale in this contribution, since we observe a coherent nature of the twinning structures; including the 1:5 boundary phase and the Z-phase. This was predicted by the theory proposed by Maury et al.[35], that is, that the Z-phase lamellas prefer to grow from twin boundaries in the 2:17 phase. However, this is not always necessarily the case, since the processing parameters play a fundamental role for the twin formation. For example, when fast cooling is applied[35] micro-twinning occurs, that is, resulting in 5–10-nm-thick twins inside the 2:17 matrix. This has also been reported by Hiraga et al.[16] and Yang et al.[19] but was not observed in our samples. Maury et al.[35] also postulate that in multicomponent phases, Fe and Cu atoms substitute for Co atoms without strong change of the dimensions, which is not true for Zr, because its atomic size is between Sm and Co. Nevertheless, Zr was predicted to be located on specific substitution sites which is indeed confirmed directly by our atomically resolved STEM-HAADF Z-contrast images, that is, specific substitution namely on the Sm1 (6c) lattice site of the SmCo₃ phase, as clearly shown by comparing Figs 2a,b.

Maury et al.[35] predicted that the Z-phase platelet formation along the basal plane is due to a significant local Zr supersaturation acting as a driving force for the nucleation.

**Micromagnetic simulations.** After obtaining the microstructural information by (S)TEM, we measured the macroscopic magnetic properties which are extracted from hysteresis loops for both samples, as shown in Table 1. The coercive field strength ($H_{cB}$) is drastically decreased from 870 kA m⁻¹ for sample 1 (lower iron content) to 250 kA m⁻¹ for sample 2 (high iron content). This results in a relatively low remanence and energy density for sample 2. For sample 1 attractive magnetic properties were achieved: $B_r$ = 1.2 T, $(BH)_{max}$ = 262 kJ m⁻³, $H_{cB}$ = 870 kA m⁻¹. We attribute the difference in the magnetic properties of these two samples to their distinguished microstructures. Since Sm₂Co₁₇ is a typical pinning-controlled magnet, we used microstructure-based micromagnetic simulations to qualitatively elucidate the domain pinning in these two samples. In order to reduce the computation cost, half of the TEM images shown in Figs 1a,c were adopted to construct the micromagnetic models with a size of 440 × 440 × 220 nm³, as shown in Supplementary Fig. 4. The initial domain wall lies in the plane parallel to the easy axis and separates two antiparallel magnetic domains (Supplementary Fig. 4). With this initial condition, micromagnetic simulations are carried out to calculate the demagnetization curves (Fig. 3a) and capture the magnetization reversal process (Fig. 3b,c). The plateaus in Fig. 3a are a result of domain wall pinning. The zigzag domain walls in Fig. 3b and c indicate the 1:5 phase and the Z-phase as pinning sites. It is obvious that sample 1 exhibits much more plateaus, thus, much more pining sites as compared to sample 2. This is in agreement with the experimental results (sample 1 has higher coercivity). It should be noted that the

micromagnetic model is only an extremely small part of the real sample. So the simulated reversal curves cannot be directly compared with the experimental curves. Nevertheless, we analyzed the detailed reversal process to reveal the underlying microstructure-related mechanism. In Figs 3b,c, P1- and P2-type sites show representative pinning sites where the 1:5 phase intersects with the Z-phase, and P1'- and P2'-type sites show representative sites only with the Z-phase. It should be mentioned that unlike the P1- and P2-type pinning sites, P1'- and P2'-type

sites are not fixed to particular positions. They move intermittently and their actual position is determined by the interplay between the cost of domain wall energy and the gain of magnetostatic energy (Supplementary Movies 1 and 2). As shown in 1i–1v of Fig. 3b, P1-type sites in sample 1 are strongly pinned until the external field reaches ~1,200 kA m$^{-1}$. But intermittent movements of domain walls occur in the P1'-type sites, resulting in lots of plateaus between 0 to ~1,000 kA m$^{-1}$ in the reversal curve of sample 1 (Fig. 3a). In contrast, in sample 2 domain walls

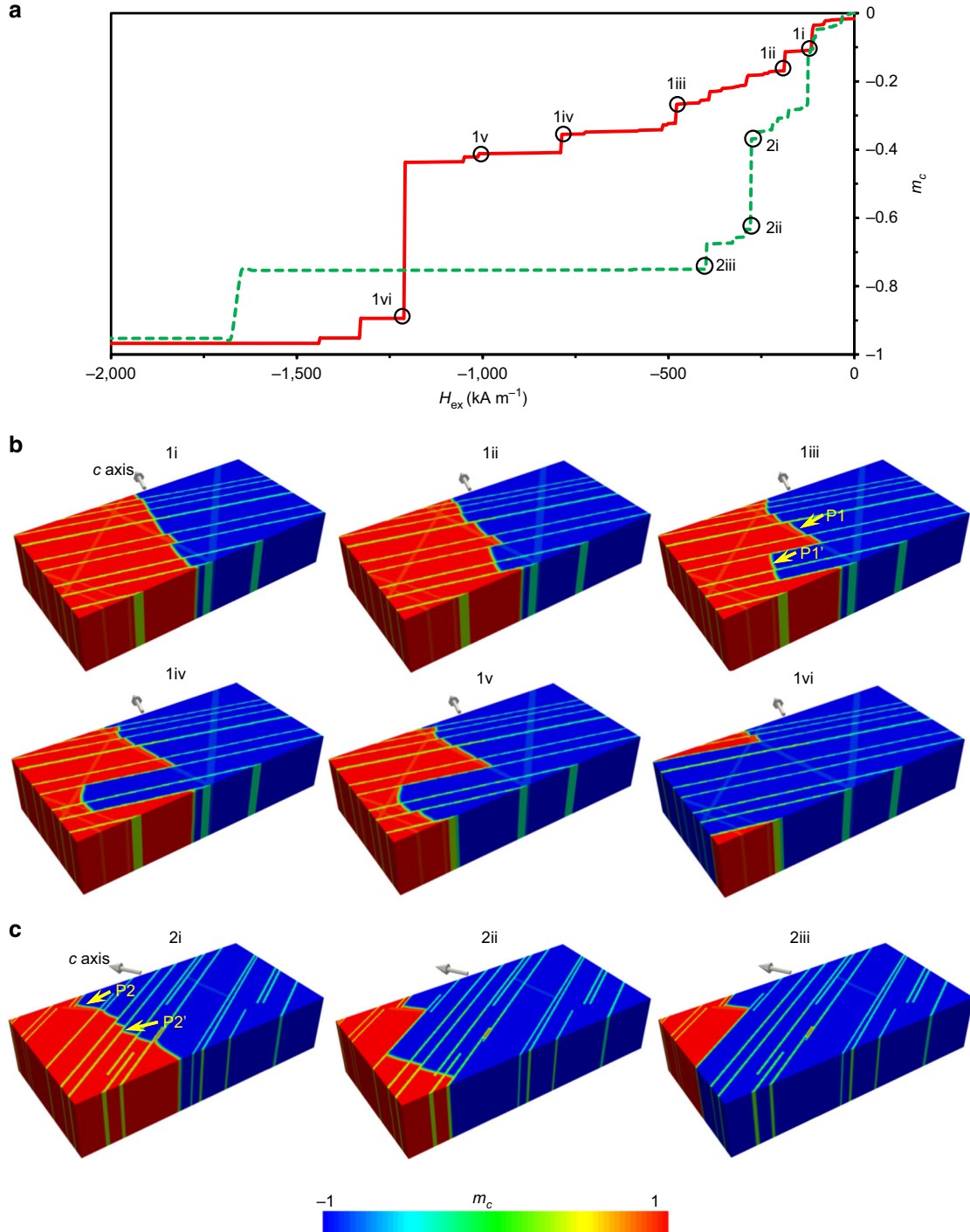

**Fig. 3** Simulation results on domain wall pinning. **a** Demagnetization curves with the *red and green lines* corresponding to samples 1 and 2, respectively. The magnetization reversal process of **b** sample 1 and **c** sample 2 at different values of applied external magnetic field marked in **a**. $m_c$ denotes the magnetization component along the easy axis. P1 and P2 show typical pinning sites, where 1:5 phase intersects with the Z-phase. P1' and P2' denote typical sites containing only the Z-phase. The *yellow arrows* denote the positions

intermittently move much faster in P2′-type sites and rapidly sweep through most of the sample at a low external field of ~400 kA m$^{-1}$ (2iii in Fig. 3c). But P2-type sites in 2i of Fig. 3c are still strongly pinned. Furthermore, the domain wall cannot be pinned by Z-phase any more when it is depinned in 1:5 phase. As shown in Fig. 3b(1v–1vi), the domain wall in 1v contains a long segment in the Z-phase, but this segment collapses immediately after it is not stabilized by the 1:5 phase (1vi). This indicates that in both samples, 1:5 phase related, P1- and P2-type sites have larger pinning strength than Z-phase related P1′- and P2′-type sites. The diamond-shaped cellular structure with a continuous 1:5 phase, i.e. more P1- and P2-type sites, is responsible for the increased number and pinning strength of the pinning sites, which are favorable for an enhanced coercivity. The Z-phase can act as pinning sites, but contributes little to the coercivity. Since the domain wall energy in the Z-phase is lower compared to the 2:17 and the 1:5 phase, the Z-phase acts as a weak attractive pinning site. In contrast, the 1:5 phase act as repulsive pinning site because of its much higher domain wall energy.

**Conclusions**. This contribution presents a detailed microstructural and chemical investigation of $Sm_2(Co,Fe,Cu,Zr)_{17}$ sintered permanent magnets with different iron content. A major objective was to use atomically resolved STEM-HAADF Z-contrast imaging in combination with micromagnetic simulations to directly determine the atomic structure of the pinning relevant phases (1:5 and Z-phase) and their magnetic behavior. Low iron content leads to superstructure type ordering of the Zr-rich platelets, which contribute to the formation of a well-developed 1:5 phase and thus a diamond-shaped cellular structure. The coercivity is dominated by the density and strength of the pinning sites in the 1:5 phase while modified by the Z-phase. Via direct atomic scale observations we demonstrate that Zr preferably replaces the Sm atoms located at the Sm1 (6c) site in the $SmCo_3$ structure yielding a modified structure with the following sum formula: $Zr_2SmCo_9$. This enables a comprehensive way of tailoring the magnetic properties, for example, coercivities since Zr favors Z-phase nucleation and controls diffusion along the Zr-rich platelets stabilizing the diamond-shaped cellular 1:5 phase. An enhanced understanding of the pinning mechanisms in $Sm_2Co_{17}$ yields a viable route to apply these thermal and chemical protocols for improved magnetic performances also to other systems than $Sm_2Co_{17}$. Further studies in this material system focusing on nanoscale spin dynamics and on the redistribution of elements like Cu and Zr on the nanoscale during different annealing programs as well as structural and chemical changes after doping by other rare-earth elements will be carried out in the near future and are beyond the scope of the present contribution.

## Methods

**Sample synthesis**. Several SmZrCoFeCu 2:17 master-alloys were melted in a vacuum furnace, crushed to coarse powders and jet milled in an AFG100 down to a particle size of ~6 μm. The compositions of these fine powders were verified by chemical analysis. The fine powders were blended in order to meet the composition (wt%) $Sm_{25}Zr_3Co_{49}Fe_{19}Cu_5$ and $Sm_{25}Zr_3Co_{45}Fe_{23}Cu_5$. The fine powders were oriented in a magnetic field of 13 kA cm$^{-1}$ and pressed isostatically at 250 MPa. The green compacts were sintered at 1190 °C, homogenized at 1,160 °C and quenched to room temperature. The samples were annealed at 870 °C and cooled slowly with a cooling rate of about 1 °C min$^{-1}$ to 400 °C.

**Electron microscopy**. A 200 kV Jeol JEM 2100 F STEM equipped with an Oxford X-max$^{80}$ EDX detector was used to determine the microstructure and chemistry on the nanometer scale. For the TEM studies, the samples were demagnetized, thinned via conventional grinding and polishing plane to a thickness of ~20 μm with the hexagonal c-axis of $Sm_2Co_{17}$ lying perpendicular to the polishing direction and finally mounted on Mo grids. Ion thinning was done in a Gatan Dual Ion Mill Model 600 using Ar$^+$ ions with an incidence angle of 15° at 5 keV. The previous

milling step was followed by two 10 min polishing steps at 13° ion incidence using 3 keV and a final step at 1.5 keV. A plasma cleaning step was performed for 2 h in a Gatan Solarus plasma cleaning system before introducing the specimen in the microscope. Annular dark-field (ADF) scanning transmission electron microscope (STEM) images and energy-dispersive X-ray (EDX) spectra were obtained using a 0.7 nm spot size, this being a compromise between spatial resolution and EDX signal (detector dead time around 10%). Quantification of EDX spectra was carried out standardless using the Cliff-Lorimer k-factor method (Supplementary Note 1). Atomic resolution images were acquired using a Jeol Atomic Resolution Microscope (ARM) 200 F equipped with a Schottky emitter and a $C_s$-probe corrector (see also Supplementary Note 1). An electron energy of 120 keV was used to reduce magnetization effects of the sample. High-angle annular dark-field (HAADF) images were acquired using the 8 C spot size setting. A 30 μm condenser aperture was inserted yielding a convergence angle of 24.6 mrad. 6 cm camera length was used for HAADF imaging corresponding to a 90 mrad inner and a 370 mrad outer HAADF detector angle. The specimen thickness according to electron energy-loss spectroscopy (EELS) was estimated to 0.4–0.7 mean free path (mfp) for sample 1 and 0.7–1.0 mfp for sample 2.

STEM-HAADF Z-contrast image simulation was carried out using the QSTEM software$^{32}$. A $C_s$ value of 1 μm, a $C_c$ value of 1 mm and an energy spread of 0.7 eV were assumed; higher order aberrations were neglected. For the convergence angle and the HAADF detector the values listed in the previous paragraph were used. Thermal diffuse scattering was not considered. For the simulation two structural modifications were used: pure $SmCo_3$ (space group R-3m) and $Zr_2SmCo_9$, which is essentially the same as the pure compound, but with the Sm1 (6c) atomic position (6c, $x = 0$, $y = 0$, $z = 0.141$) replaced with Zr. STEM-HAADF images were Wiener filtered for noise reduction$^{36}$. Selected STEM-HAADF images were filtered using a 5–15 component principal component analysis for improved noise reduction$^{37}$.

**Micromagnetic simulations**. In the micromagnetic simulations of the magnetization reversal process of the model sample 1 and 2, microstructure-oriented models were discretized by cubic meshes with a size of 1 nm. The Landau–Lifshitz–Gilbert equation at each node was solved by the 3D NIST OOMMF software. The magnetocrystalline anisotropy values of the 1:5 phase, the 2:17 phase, and the Z-phase are taken from the literature$^{29}$ as 12.1 MJ m$^{-3}$, 3.9 MJ m$^{-3}$, and 2.1 MJ m$^{-3}$, respectively. The magnitude of the saturation magnetization of 1:5 phase, 2:17 phase and Z-phase are taken from the literature$^{29}$ as 1.1, 1.23, and 0.39 T, respectively. The exchange constant of the 1:5 phase, the 2:17 phase, and the Z-phase is estimated from the literature$^{29}$ as 15.1, 19.6, and 0.48 pJ m$^{-1}$, respectively. Supplementary Fig. 4 shows the three-dimensional(3D) micromagnetic model of samples 1 and 2. The models (440 × 440 × 220 nm$^3$) only consider the upper half of the microstructure shown in the TEM images (Fig. 1), in order to lower the computation cost. The stripes denote the 1:5 phase and Z-phase. An initial 180° domain wall along the easy axis is set to study the domain wall pinning effect. The external field is applied antiparallel to the arrows (c-axis) shown in Supplementary Fig. 4. The simulated magnetization reversal process is shown in Supplementary Movies 1 and 2.

**First principle calculations**. First-principles calculations based on density functional theory were performed by using the Vienna ab initio simulation package. The exchange correlation energy was calculated within the generalized gradient approximation of the Perdew–Burke–Ernzerhof (PBE) form. The cutoff energies for the plane wave basis set to expand the Kohn-Sham orbitals were 500 eV for all calculations. The energies through simulation mentioned in this work are the energies after structural relaxation. Γ centered 9 × 9 × 2 and 15 × 15 × 3 K-point mesh within Monkhorst-Pack scheme was used for the Brillouin zone integration for structural relaxation and energy calculation, respectively. The structural relaxation was done until the forces were smaller than 2 meV Å$^{-1}$. Supplementary Fig. 5 shows the calculated energies of $SmCo_3$ with different Sm sites replaced by Zr atoms.

**Data availability**. The data files that support the findings of this study are available from the corresponding author upon reasonable request.

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

## Acknowledgements

We acknowledge financial support from the Hessen State Ministry of Higher Education, Research and the Arts via LOEWE RESPONSE. The transmission electron microscopes used in this work were partially funded by the German Research Foundation (DFG/INST163/2951). The authors thank Maximilian Trapp for TEM sample preparation and acknowledge the use of Lichtenberg High Performance Computer of TU Darmstadt.

## Author contributions

M.D. and L.M.-L. designed and conducted the (scanning) transmission electron microscopy experiments and simulations, analyzed the acquired data and wrote the corresponding manuscript part. K.U., M.L. and M.K. synthesized the $Sm_2Co_{17}$ samples and obtained the corresponding magnetic data. H.-J.K. commented on the TEM imaging. M.Y. and B.X. performed the micromagnetic simulations, density functional calculations and wrote the corresponding manuscript part. O.G., L.M.-L. and M.K. motivated this study. M.D. and M.Y. contributed equally to the joint experimental and theoretical work. L.M.-L. coordinated and led this investigation. All authors discussed extensively the results and commented on the manuscript.

## Additional information

**Competing interests:** The authors declare no competing financial interests.

