## [Peer Review file · Nature Communications]

Reviewers' comments:

Reviewer #1 (Remarks to the Author):

The paper by Duerrschabel et al. deals with the development and understanding of pinning-type Sm-Co magnets. The topic is of great importance in magnetism and suitable for Nature Communications. The experimental part of the paper is of a very high quality, including the results of Table 1, but I have some concerns about the physical picture advocated in the paper and about some aspects of the presentation of the manuscript. My comments are as follows:

(i) The presentation is not ideal in several regards: (a) The title is rather long, and "Unraveling ..." sound fluffy and is not supported by the paper (see below); (b) The first sentence of the main text starts with the phrase "Super-powerful pinning-controlled PM ...", which sounds like puffery and contains an unnecessary acronym; (c) Parts of the paper give a rather specialized impression (p. 4, p. 6) and may be difficult to follow except for experts; (d) How do the present energy products compare to those of other top Sm-Co grades?; (e) The first author does not have an affiliation.

(ii) To motivate their work, the authors claim (line 58) that Ref. 10 concludes that the Zr-rich platelets "do not act as pinning centers". In fact, Ref. 10 states very clearly that the platelets "may act as pinning centers" (8th line of conclusions).

(iii) The main point of Ref. 10 is that domain walls get curved and are no longer parallel to field and c-axis, becoming more or less parallel to the 1:5 regions. In terms of Fig. 3 in the reference, the difference is that between the top and bottom rows, the top row typical of small fields and bottom row physically realized near coercivity. As a consequence, the main pinning contribution comes from the domain wall at the 1:5 cell boundary and is merely modified by cross-cutting platelets and other imperfections. Without this boundary, the coercivity would essentially be zero, but this point is essentially swept under the rug by the authors.

(iv) The micromagnetic simulations of Fig. 3(b) in the present manuscript confirm rather than contradict the picture elaborated in Ref. 10. In small fields (1i, 1ii), the wall is nearly parallel to the c-axis, whereas in fields approaching the coercivity (1iv, 1v), the simulation looks like Fig. 3(c) in Ref. 10. Of course, if one puts a domain-wall into the middle of a cell, then the wall exploits the reduced domain-wall energy in the platelet phase for preliminary readjustment, but this does not control the coercivity. Comparison of (1v) and (1vi) clearly indicates that the coercivity is controlled by the domain wall's jump over the 1:5 region, resisted by the 1:5 region but driven by the external field and supported (not resisted) by the grain-boundary phase. (The domain wall of 1v contains a long segment in the platelet region, but this segment would collapse immediately if it was not stabilized by the 1:5 phase. In fact, since the collapse is energetically favorable, unlike the overcoming of the 1:5 pinning barrier, its coercivity contribution is not only small but actually negative.)

The response to (i) should be straightforward. Points (ii-iv) are related and indicate an overselling of otherwise very good results, with the effect of producing a poorly supported and probably incorrect physical picture. I recommend that the authors present their paper as substantial progress in the investigation of a longstanding but only partially solved important problem. Maintaining the view that the coercivity is dominated rather than modified by the platelet phase would require major changes in the paper and may go nowhere.

In summary, I recommend the publication of this manuscript after mandatory but not necessarily extensive revision.

Reviewer #2 (Remarks to the Author):

Publication Peer Review

Manuscripts judged to be of potential interest to our readership are sent for formal review, typically to two or three reviewers, but sometimes more if special advice is needed (for example on statistics or a particular technique). The editors then make a decision based on the reviewers' advice, from among several possibilities:

- **Accept, with or without editorial revisions**
- Invite the authors to revise their manuscript to address specific concerns before a final decision is reached
- Reject, but indicate to the authors that further work might justify a resubmission
- Reject outright, typically on grounds of specialist interest, lack of novelty, insufficient conceptual advance or major technical and/or interpretational problems

Reviewer's comments are in blue text.

Recommend Accept with or without changes or additions discussed below.

- Who will be interested in reading the paper, and why?
Members of the magnetics community including magnet manufacturers and users of magnets for demanding applications and the research laboratories within companies and at universities.
- What are the main claims of the paper and how significant are they?
Objective of the paper is to explain the structural differences between normal and elevated iron content SmCo 2:17 magnets. Premise is that understanding these differences might lead to methods for improvements in magnetic properties, specifically energy product and resistance to demagnetization.
- Is the paper likely to be one of the five most significant papers published in the discipline this year?
The focus of recent magnetics research has been on NdFeB magnets. However, this paper will be among the top five in high performance magnets as SmCo offers a superb alternative to NdFeB in high temperature demanding applications.
- How does the paper stand out from others in its field?
The quality of the micrography is outstanding and the interpretation of the structure is superior to earlier publications.
- Are the claims novel? If not, which published papers compromise novelty?
Discussion of structure and mechanisms of coercivity (domain reversal) builds upon previous published material in a substantive way.
- Are the claims convincing? If not, what further evidence is needed?
Well-documented and convincing.
- Are there other experiments or work that would strengthen the paper further?
I was disappointed in the absence of suggestions for follow-on research such as composition optimization especially regarding copper and zirconium content or the introduction of alternate

rare earth elements to stress the lattice causing structural modification.

- How much would further work improve it, and how difficult would this be? Would it take a long time?
This paper is complete as it stands. Further work would be beyond the scope of this paper.
- Are the claims appropriately discussed in the context of previous literature?
There are in excess of 450 prior papers on SmCo 2:17. Appropriate references have been selected from this large body of literature.
- If the manuscript is unacceptable, is the study sufficiently promising to encourage the authors to resubmit?
n/a
- If the manuscript is unacceptable but promising, what specific work is needed to make it acceptable?
n/a
- Is the manuscript clearly written?
Yes, excellent English with almost faultless continuity of idea flow. It is highly technical and likely to appeal most to the research community.
- If not, how could it be made more clear or accessible to non-specialists?
I believe it would be inappropriate to attempt to “water-it-down” for general consumption – its value lies in the scientific complexity of the content: STEM-HAADF, QSTEM, DFT, domain reversal modeling, etc.
- Would readers outside the discipline benefit from a schematic of the main result to accompany publication?
The abstract is a bit misleading and I would suggest some modifications as indicated below (at end of Q&A section. These clarifications should provide a benefit to “casual” readers.
- Could the manuscript be shortened? (Because of pressure on space in our printed pages we aim to publish manuscripts as short as is consistent with a persuasive message.)
This paper represents a great deal of work on the part of the authors and it is tempting to include everything in the paper. It could probably be shortened by 25% and retain the main ideas/content, but I would be hesitant to suggest this as its all good reading.
- Should the authors be asked to provide supplementary methods or data to accompany the paper online? (Such data might include source code for modelling studies, detailed experimental protocols or mathematical derivations.)
The “Methods” section starting on page ten suffices.
- Have the authors done themselves justice without overselling their claims?
Generally yes, though see suggested changes to the abstract (below).
- Have they been fair in their treatment of previous literature?
Yes though there are some additional references related to previous efforts at using higher iron content. This can be excused due to this paper’s focus on explaining the microstructure rather

than property optimization and to the large existing body of science.

- Have they provided sufficient methodological detail that the experiments could be reproduced?
Yes.
- Is the statistical analysis of the data sound, and does it conform to the journal's guidelines?
Of necessity, there are a limited number of samples but the results are clear and not subject to misinterpretation.
- Are the reagents generally available?
Raw materials are generally available, though processing methods require equipment that is less common outside the manufacturing community, e.g., fluid bed jet mill AFG-100.
- Are there any special ethical concerns arising from the use of human or other animal subjects?
n/a

ABSTRACT

Suggested changes indicated in blue font and by strikethrough formatting

A higher saturation magnetization obtained by increased iron content is essential for yielding ~~even better magnetic performances~~ higher energy product in rare-earth (RE) Sm₂Co₁₇-type 'pinning controlled' permanent magnets (PM).¹⁻⁴ ~~These are SmCo is~~ of importance for high-temperature industrial applications due to their high intrinsic corrosion resistance and temperature stability.⁵⁻⁹ Based on a unique nanostructure and -chemical modification route using Fe, Cu, and Zr dopants¹⁰⁻¹⁴ we produced model magnets with increased iron content to precisely tailor the atomic structure and its domain wall pinning behavior. The iron content controls the phase formation of a diamond-shaped cellular structure that dominates the density and strength of the pinning sites.¹⁵ By a combination of ultra-high resolution experimental and theoretical methods, we were able to reveal the atomic structure of the single phases present and to establish a direct correlation to the macroscopic magnetic properties. ~~The knowledge is used~~ With further development, this knowledge may be applied to produce samarium cobalt PM with higher magnetic performances.

Rationale for the changes

Iron is introduced to increase magnetic saturation. With development of a "good" microstructure resulting in domain pinning, this produces both a high Br and larger energy product. Normally, the higher iron content, above about 20 weight percent, results in a reduction of pinning starting with reduced H_{cj} (especially above about 19 weight percent iron) and then reduced Br (above about 20 weight percent iron). Without additional improvement in properties the tested 23 weight percent would be of limited commercial value.

ADDITIONAL SUBSTANTIVE REFERENCES

A substantial amount of research has been conducted on higher iron content and changes to processing and structure of SmCo magnets. Inclusion of some or all of these references would lend additional credence to the excellent references already included in the publication – listed in alphabetic order based on title.

- 2-17 type RE-TM magnets with improved magnetic properties - Liu and Ray – REPM - 1989
- Developments in Magnetic Properties of Resin Bonded Sm₂TM₁₇ Type Magnet - Shimoda et al – REPM – 1981

- Effect of Fe on the structure and magnetic properties of SmCoCuFeZr melt-spun ribbons - Sun et al - Materials Science and Engineering B 157 (2009) 72–76
- Extending the limits of the Sm₂Co₁₇ System - Maybury et al - WMM - 2016
- Influence of Fe content on magnetic properties of high temperature rare earth permanent magnets Sm(CoFeCuZr)_{7.5} - Xu et al - Rare Metal Materials and Engineering Volume 37, Issue 3, March 2008
- Influence of iron content on the magnetic properties of Sm(Co,Fe,Cu,Zr)_z magnets with high coercive force - Lan et al – REPM – 1983
- Melt Spinning Process and Its Effect on the Magnetic Properties and Structure of Sm(Co,Fe-Cu,Zr)_z Melt-Spun Ribbons - Liu et al – IEEE – 2015
- Microstructure and properties of step aged rare earth alloy magnets - Mishra et al - J. Appl. Phys. 52 (3), March 1981
- Microstructure of Sm₂Co₁₇ magnets and its influence on coercivity - Li et al – Trans. Nonferrous Met. Soc. China Vol. 14 No. 4 – August 2004
- Sm₂(Co,Fe,Cu,Zr)₁₇ Magnets with Higher Fe Content - Liu and Ray – IEEE – 1989
- SmCo (2-17 type) magnets with high contents of Fe and light rare earths - Huang, Zheng, Wallace - J. Appl. Phys. 75 (10), 15 May 1994

FINAL COMMENT

On page 5, in the start of the discussion of HAADF, it is not clear why the two model compositions were selected. A brief explanation would be useful as to what led to selecting the two specific QSTEM models SmCo₃ and Zr₂SmCo₉.

Respectfully submitted,
 Steve Constantinides
 December 8, 2016

Reviewer #3 (Remarks to the Author):

This paper studies the effect of Fe content in Sm₂Co₁₇-type permanent magnets on the morphology and the coercivity, in combination of the experimental investigations and the theoretical simulations and calculations. Some interesting new results are reported, for instance, the preferable substitution sites by Zr atoms in the SmCo₃ structure. However it is my opinion that this paper is still premature to be published in Nature Communication in view of the following considerations:

1. To attribute the magnetic performance of the magnets to the Fe concentrate only is not evident enough. It is well known that the cellular-shaped microstructure with pinning centers is formed due not only to the Fe addition but also to the Cu and Zr addition in a suitable amount, as well as a proper heat treatment process. While claimed by the authors that a relationship between atomic structure and the performance has been unraveled, the correlation between the Fig. 1 (microstructure) and Fig. 2 (atomic structure) is not well established.
2. No quantitative results on the atomic concentration in a reasonable range and the properties can be found in the paper, except two samples of "low Fe content sample" with 19wt% and "high Fe content sample" with 23wt%. Generally speaking, a comparison between two points in any experiments is not convincing enough. On the other hand, there has been a trend in this research area that more Fe addition is pursued in order to boost the performance.
3. Although claimed by the authors in the introduction that "the knowledge is used to produce SmCo PM with higher performances", no specific results are given in the paper.

Reviewer #1 (Remarks to the Author):

The paper by Duerrschabel *et al.* deals with the development and understanding of pinning-type Sm-Co magnets. The topic is of great importance in magnetism and suitable for Nature Communications. The experimental part of the paper is of a very high quality, including the results of Table 1, but I have some concerns about the physical picture advocated in the paper and about some aspects of the presentation of the manuscript. My comments are as follows:

- (i) The presentation is not ideal in several regards: (a) The title is rather long, and "Unraveling ..." sound fluffy and is not supported by the paper (see below); (b) The first sentence of the main text starts with the phrase "Super-powerful pinning-controlled PM ...", which sounds like puffery and contains an unnecessary acronym; (c) Parts of the paper give a rather specialized impression (p. 4, p. 6) and may be difficult to follow except for experts; (d) How do the present energy products compare to those of other top Sm-Co grades?; (e) The first author does not have an affiliation.

(a)-(c) The title and the abstract have been modified accordingly. The changes in the revised manuscript are highlighted in yellow. Although some parts of the paper may seem rather specialized we believe that the content is still accessible to the non-specialized reader without losing its scientific complexity.

(d) A reference has been inserted containing values of commercial magnets for comparison. (Reference 21, Table 1).

(e) The missing affiliation has been added.

- (ii) To motivate their work, the authors claim (line 58) that Ref. 10 concludes that the Zr-rich platelets "do not act as pinning centers". In fact, Ref. 10 states very clearly that the platelets "may act as pinning centers" (8th line of conclusions).

The sentence has been corrected and highlighted in yellow.

- (iii) The main point of Ref. 10 is that domain walls get curved and are no longer parallel to field and c-axis, becoming more or less parallel to the 1:5 regions. In terms of Fig. 3 in the reference, the difference is that between the top and bottom rows, the top row typical of small fields and bottom row physically realized near coercivity. As a consequence, the main pinning contribution comes from the domain wall at the 1:5 cell boundary and is

merely modified by cross-cutting platelets and other imperfections. Without this boundary, the coercivity would essentially be zero, but this point is essentially swept under the rug by the authors.

The section was modified for clarification and highlighted in yellow.

- (iv) The micromagnetic simulations of Fig. 3(b) in the present manuscript confirm rather than contradict the picture elaborated in Ref. 10. In small fields (1i, 1ii), the wall is nearly parallel to the c-axis, whereas in fields approaching the coercivity (1iv, 1v), the simulation looks like Fig. 3(c) in Ref. 10. Of course, if one puts a domain-wall into the middle of a cell, then the wall exploits the reduced domain-wall energy in the platelet phase for preliminary readjustment, but this does not control the coercivity. Comparison of (1v) and (1vi) clearly indicates that the coercivity is controlled by the domain wall's jump over the 1:5 region, resisted by the 1:5 region but driven by the external field and supported (not resisted) by the grain-boundary phase. (The domain wall of 1v contains a long segment in the platelet region, but this segment would collapse immediately if it was not stabilized by the 1:5 phase. In fact, since the collapse is energetically favorable, unlike the overcoming of the 1:5 pinning barrier, its coercivity contribution is not only small but actually negative.)

Thanks for the reviewer's comments. Our simulation result really confirms the picture elaborated in Ref. 10. The simulated curved domain wall configuration in Fig. 3 is also consistent with the physical model proposed in Ref. 10 (Fig. 3c). In our simulation, the pinning sites in the 1:5 phase are fixed until the external magnetic field (H_{ex}) is increased to a sufficiently high value. In contrast, the pinning sites in the platelet Z-phase move intermittently even at a low H_{ex} (Fig. 3b 1i-1v, Fig. 3c 2i-2iii), and cannot pin the domain wall any more when the domain wall is depinned in the 1:5 phase (Fig. 3b 1v-1vi). These indicate that the 1:5 phase touching the curved domain walls dominates the coercivity enhancement. The Z-phase platelet can act as pinning sites, but contribute little to the coercivity. Their main contribution is favoring the formation of cell boundaries. By maintaining the viewpoint that "The coercivity is dominated by the density and strength of the pinning sites in 1:5 phase while modified by the Z-phase which can act as pinning site but mainly contribute to the formation of well-developed 1:5 phase and thus a diamond-shaped cellular structure", we rephrase the related sentences in the abstract, the discussion in the simulation part, and the conclusion part, as highlighted in yellow in the manuscript.

Reviewer #2 (Remarks to the Author):

SEE UPLOADED FILE FOR FORMATTED REVIEW.

Publication Peer Review

Manuscripts judged to be of potential interest to our readership are sent for formal review, typically to two or three reviewers, but sometimes more if special advice is needed (for example on statistics or a particular technique). The editors then make a decision based on the reviewers' advice, from among several possibilities:

- **Accept, with or without editorial revisions**
- Invite the authors to revise their manuscript to address specific concerns before a final decision is reached
- Reject, but indicate to the authors that further work might justify a resubmission
- Reject outright, typically on grounds of specialist interest, lack of novelty, insufficient conceptual advance or major technical and/or interpretational problems

Reviewer's comments are in blue text.

Recommend Accept with or without changes or additions discussed below.

1. Who will be interested in reading the paper, and why?

Members of the magnetism community including magnet manufacturers and users of magnets for demanding applications and the research laboratories within companies and at universities.

2. What are the main claims of the paper and how significant are they?

Objective of the paper is to explain the structural differences between normal and elevated iron content SmCo 2:17 magnets. Premise is that understanding these differences might lead to methods for improvements in magnetic properties, specifically energy product and resistance to demagnetization.

3. Is the paper likely to be one of the five most significant papers published in the discipline this year?

The focus of recent magnetism research has been on NdFeB magnets. However, this paper will be among the top five in high performance magnets as SmCo offers a superb alternative to NdFeB in high temperature demanding applications.

4. How does the paper stand out from others in its field?

The quality of the micrograph is outstanding and the interpretation of the structure is superior to earlier publications.

5. Are the claims novel? If not, which published papers compromise novelty?

Discussion of structure and mechanisms of coercivity (domain reversal) builds upon previous published material in a substantive way.

6. Are the claims convincing? If not, what further evidence is needed?

Well-documented and convincing.

7. Are there other experiments or work that would strengthen the paper further?

I was disappointed in the absence of suggestions for follow-on research such as composition optimization especially regarding copper and zirconium content or the introduction of alternate rare earth elements to stress the lattice causing structural modification.

We have added a sentence highlighted in yellow at the end of the manuscript regarding this point.

8. How much would further work improve it, and how difficult would this be? Would it take a long time?

This paper is complete as it stands. Further work would be beyond the scope of this paper.

9. Are the claims appropriately discussed in the context of previous literature?

There are in excess of 450 prior papers on SmCo 2:17. Appropriate references have been selected from this large body of literature.

10. If the manuscript is unacceptable, is the study sufficiently promising to encourage the authors to resubmit?

n/a

11. If the manuscript is unacceptable but promising, what specific work is needed to make it acceptable?

n/a

12. Is the manuscript clearly written?

Yes, excellent English with almost faultless continuity of idea flow. It is highly technical and likely to appeal most to the research community.

13. If not, how could it be made more clear or accessible to non-specialists?

I believe it would be inappropriate to attempt to “water-it-down” for general consumption – its value lies in the scientific complexity of the content: STEM-HAADF, QSTEM, DFT, domain reversal modeling, etc.

14. Would readers outside the discipline benefit from a schematic of the main result to accompany publication?

The abstract is a bit misleading and I would suggest some modifications as indicated below (at end of Q&A section. These clarifications should provide a benefit to “casual” readers.

15. Could the manuscript be shortened? (Because of pressure on space in our printed pages we aim to publish manuscripts as short as is consistent with a persuasive message.)

This paper represents a great deal of work on the part of the authors and it is tempting to include everything in the paper. It could probably be shortened by 25% and retain the main ideas/content, but I would be hesitant to suggest this as its all good reading.

16. Should the authors be asked to provide supplementary methods or data to accompany the paper online? (Such data might include source code for modelling studies, detailed experimental protocols or mathematical derivations.)

The “Methods” section starting on page ten suffices.

17. Have the authors done themselves justice without overselling their claims?

Generally yes, though see suggested changes to the abstract (below).

18. Have they been fair in their treatment of previous literature?

Yes though there are some additional references related to previous efforts at using higher iron content. This can be excused due to this paper's focus on explaining the microstructure rather than property optimization and to the large existing body of science.

19. Have they provided sufficient methodological detail that the experiments could be reproduced?

Yes.

20. Is the statistical analysis of the data sound, and does it conform to the journal's guidelines?

Of necessity, there are a limited number of samples but the results are clear and not subject to misinterpretation.

21. Are the reagents generally available?

Raw materials are generally available, though processing methods require equipment that is less common outside the manufacturing community, e.g., fluid bed jet mill AFG-100.

22. Are there any special ethical concerns arising from the use of human or other animal subjects?

n/a

ABSTRACT

Suggested changes indicated in blue font and by strikethrough formatting

A higher saturation magnetization obtained by increased iron content is essential ~~for yielding even better magnetic performances~~ higher energy product in rare-earth (RE) Sm₂Co₁₇-type 'pinning controlled' permanent magnets (PM). 1–4 ~~These are SmCo~~ is of importance for high-temperature industrial applications due to their high intrinsic corrosion resistance and temperature stability. 5–9 Based on a unique nanostructure and –chemical modification route using Fe, Cu, and Zr dopants 10–14 we produced model magnets with increased iron content to precisely tailor the atomic structure and its domain wall pinning behavior. The iron content controls the phase formation of a diamond-shaped cellular structure that dominates the density and strength of the pinning sites. 15 By a combination of ultra-high resolution experimental and theoretical methods, we were able to reveal the atomic structure of the single phases present and to establish a direct correlation to the macroscopic magnetic properties. ~~The knowledge is used~~ With further development, this knowledge may be applied to produce samarium cobalt PM with higher magnetic performances.

The abstract has been corrected as suggested and changes are highlighted in yellow.

Rationale for the changes

Iron is introduced to increase magnetic saturation. With development of a "good" microstructure resulting in domain pinning, this produces both a high Br and larger energy product. Normally, the higher iron content, above about 20 weight percent, results in a reduction of pinning starting with reduced Hcj (especially above about 19 weight percent

iron) and then reduced Br (above about 20 weight percent iron). Without additional improvement in properties the tested 23 weight percent would be of limited commercial value.

ADDITIONAL SUBSTANTIVE REFERENCES

A substantial amount of research has been conducted on higher iron content and changes to processing and structure of SmCo magnets. Inclusion of some or all of these references would lend additional credence to the excellent references already included in the publication – listed in alphabetic order based on title.

- 2-17 type RE-TM magnets with improved magnetic properties - Liu and Ray – REPM – 1989
- Developments in Magnetic Properties of Resin Bonded Sm₂TM₁₇ Type Magnet - Shimoda et al – REPM – 1981
- Effect of Fe on the structure and magnetic properties of SmCoCuFeZr melt-spun ribbons - Sun et al - Materials Science and Engineering B 157 (2009) 72–76
- Extending the limits of the Sm₂Co₁₇ System - Maybury et al - WMM – 2016
- Influence of Fe content on magnetic properties of high temperature rare earth permanent magnets Sm(CoFeCuZr)_{7.5} - Xu et al - Rare Metal Materials and Engineering Volume 37, Issue 3, March 2008
- Influence of iron content on the magnetic properties of Sm(Co,Fe,Cu,Zr)_z magnets with high coercive force - Lan et al – REPM – 1983
- Melt Spinning Process and Its Effect on the Magnetic Properties and Structure of Sm(Co,Fe-Cu,Zr)_z Melt-Spun Ribbons - Liu et al – IEEE – 2015
- Microstructure and properties of step aged rare earth alloy magnets - Mishra et al - J. Appl. Phys. 52 (3), March 1981
- Microstructure of Sm₂Co₁₇ magnets and its influence on coercivity - Li et al – Trans. Nonferrous Met. Soc. China Vol. 14 No. 4 – August 2004
- Sm₂(Co,Fe,Cu,Zr)₁₇ Magnets with Higher Fe Content - Liu and Ray – IEEE – 1989
- SmCo (2-17 type) magnets with high contents of Fe and light rare earths - Huang, Zheng, Wallace - J. Appl. Phys. 75 (10), 15 May 1994

The suggested references highlighted in yellow (above) have been introduced into the manuscript, except for the oldest REPM conference proceedings articles that are not available to us and probably also would not be for many readers (three articles).

FINAL COMMENT

On page 5, in the start of the discussion of HAADF, it is not clear why the two model compositions were selected. A brief explanation would be useful as to what led to selecting the two specific QSTEM models SmCo₃ and Zr₂SmCo₉.

A sentence explaining the requested issue was introduced and highlighted in yellow.

Reviewer #3 (Remarks to the Author):

This paper studies the effect of Fe content in Sm₂Co₁₇ –type permanent magnets on the morphology and the coercivity, in combination of the experimental investigations and the theoretical simulations and calculations. Some interesting new results are reported, for instance, the preferable substitution sites by Zr atoms in the SmCo₃ structure. However it is my opinion that this paper is still premature to be published in Nature Communication in view of the following considerations:

1. To attribute the magnetic performance of the magnets to the Fe concentrate only is not evident enough. It is well known that the cellular-shaped microstructure with pinning centers is formed due not only to the Fe addition but also to the Cu and Zr addition in a suitable amount, as well as a proper heat treatment process. While claimed by the authors that a relationship between atomic structure and the performance has been unraveled, the correlation between the Fig. 1 (microstructure) and Fig. 2 (atomic structure) is not well established.

The title and the abstract have been modified to avoid misinterpretations. The main objective of this contribution is to explain the structural differences between normal and elevated iron content SmCo 2:17 magnets rather than on focusing on property optimization. Since the profound understanding of these differences is likely to lead to methods for improvements in magnetic properties, specifically energy product and resistance to demagnetization, we believe that this publication fills a much needed and longtime open gap in the literature towards obtaining this goal.

2. No quantitative results on the atomic concentration in a reasonable range and the properties can be found in the paper, except two samples of “low Fe content sample” with 19wt% and “high Fe content sample” with 23wt%. Generally speaking, a comparison between two points in any experiments is not convincing enough. On the other hand, there has been a trend in this research area that more Fe addition is pursued in order to boost the performance.

Even though we had to restrict our elaborate investigations to two samples, a clear and fundamental difference in the corresponding structural and magnetic properties became obvious. As also mentioned by reviewer 2: “of necessity, there are a limited number of samples but the results are clear and not subject to misinterpretation” –maybe not surprisingly we would fully agree with his point of view. Additional references dealing with the trend mentioned by referee 2 have been introduced and discussed carefully in the revised manuscript.

3. Although claimed by the authors in the introduction that “the knowledge is used to produce SmCo PM with higher performances”, no specific results are given in the paper.

The sentence has been modified to avoid misinterpretations. The actual production of such magnets would be beyond the scope of this paper.

REVIEWERS' COMMENTS:

Reviewer #1 (Remarks to the Author):

The authors have properly addressed my concerns and, in my opinion, those by the other referees.

A minor exception is the style of the paper, especially the yellow changes on the first page, which look overspecialized and are generally a little bit below Nature level. For example, the modified lines in the abstract contain the word "high(er)" four times, and the acronym "PM" is used five times on p. 1, including the abstract. Personally, I dislike this acronym, because in magnetism, it means both "paramagnetism, paramagnet" and "permanent magnetism, permanent magnet". The acronym "RE" is used in the abstract but nowhere else in the paper.

A style-related question with potential scientific implications is the use of the words "would" in the first sentence (line 38) and "candidate" on p. 41. These words are frequently used in the literature to characterize research that has no direct experimental or industrial implications but may become practically relevant in the future. In fact, Sm-Co is an important high-temperature material even today, much more than a "candidate". The direct relevance to real materials is a strength of the present paper and should be properly exploited. (I recently heard the argument that magnetic oxides with trivially small net magnetization M and high coercivity $\sim 2K/M$ are candidates for permanent magnets. This is a euphemism for never becoming a meaningful material, because the energy product of permanent magnets scales as M^2 .)

REVIEWERS' COMMENTS:

Reviewer #1 (Remarks to the Author):

The authors have properly addressed my concerns and, in my opinion, those by the other referees.

A minor exception is the style of the paper, especially the yellow changes on the first page, which look overspecialized and are generally a little bit below Nature level. For example, the modified lines in the abstract contain the word "high(er)" four times, and the acronym "PM" is used five times on p. 1, including the abstract. Personally, I dislike this acronym, because in magnetism, it means both "paramagnetism, paramagnet" and "permanent magnetism, permanent magnet". The acronym "RE" is used in the abstract but nowhere else in the paper.

We have removed the acronym PM and changed the abstract accordingly.

A style-related question with potential scientific implications is the use of the words "would" in the

first sentence (line 38) and "candidate" on p. 41. These words are frequently used in the literature to characterize research that has no direct experimental or industrial implications but may become practically relevant in the future. In fact, Sm-Co is an important high-temperature material even today, much more than a "candidate". The direct relevance to real materials is a strength of the present paper and should be properly exploited. (I recently heard the argument that magnetic oxides with trivially small net magnetization M and high coercivity $\sim 2K/M$ are candidates for permanent magnets. This is a euphemism for never becoming a meaningful material, because the energy product of permanent magnets scales as M^2 .)

We have removed the use of "would" and "candidate" in the first sentence. We strongly agree with the reviewer, Sm-Co is indeed a successful industrial material. We have modified the text accordingly.